# Neurovascular Impairment and Therapeutic Strategies in Diabetic Retinopathy

**DOI:** 10.3390/ijerph19010439

**Published:** 2021-12-31

**Authors:** Toshiyuki Oshitari

**Affiliations:** 1Department of Ophthalmology and Visual Science, Chiba University Graduate School of Medicine, Inohana 1-8-1, Chuo-ku, Chiba 260-8670, Japan; Tarii@aol.com; Tel./Fax: +81-43-226-2124 or +81-43-224-4162; 2Department of Ophthalmology, International University of Health and Welfare School of Medicine, 4-3 Kozunomori, Narita 286-8686, Japan

**Keywords:** neurovascular unit, polyol pathway, oxidative stress, advanced glycation-end product, neuroprotection, vasoprotection, diabetic retinopathy

## Abstract

Diabetic retinopathy has recently been defined as a highly specific neurovascular complication of diabetes. The chronic progression of the impairment of the interdependence of neurovascular units (NVUs) is associated with the pathogenesis of diabetic retinopathy. The NVUs consist of neurons, glial cells, and vascular cells, and the interdependent relationships between these cells are disturbed under diabetic conditions. Clinicians should understand and update the current knowledge of the neurovascular impairments in diabetic retinopathy. Above all, neuronal cell death is an irreversible change, and it is directly related to vision loss in patients with diabetic retinopathy. Thus, neuroprotective and vasoprotective therapies for diabetic retinopathy must be established. Understanding the physiological and pathological interdependence of the NVUs is helpful in establishing neuroprotective and vasoprotective therapies for diabetic retinopathy. This review focuses on the pathogenesis of the neurovascular impairments and introduces possible neurovascular protective therapies for diabetic retinopathy.

## 1. Introduction

According to the ninth edition of the International Diabetes Federation diabetes atlas, the estimated prevalence of diabetes will be 10.2% (578 million) by 2030 and 10.9% (700 million) by 2045 in the world [1]. Diabetic retinopathy is a major complication in diabetic patients which causes vision loss. After 20 years of diabetes, nearly all patients with type 1 and 60% of patients with type 2 diabetes develop diabetic retinopathy [2]. The vision-threatening stages of diabetic retinopathy are clinically significant diabetic macular edema (DME) and proliferative diabetic retinopathy including tractional retinal detachment and vitreous hemorrhages. The META-EYE study reported that among 22,896 diabetic patients, 10.2% had DME and/or proliferative diabetic retinopathy [3]. The Japan Diabetes Clinical Data Management Study Group reported in 2014 that 23.5% of type 2 diabetic patients had diabetic retinopathy. Because many younger patients have poor glycemic control, the prevalence of diabetic retinopathy in younger patients may increase in the future in Japan [4]. Worldwide, 95 million diabetic patients have diabetic retinopathy, and the growing number of diabetic patients will cause an increase in the number of patients with diabetic retinopathy in the future [5].

The Early Treatment Diabetic Retinopathy Study (ETDRS) has classified the stages of diabetic retinopathy based on vascular changes including dot/blot hemorrhages, hard/soft exudates, intraretinal microvascular abnormalities, and neovascularization [6]. An increase in the vascular permeability is related to the development of DME, and the ETDRS study reported that if the edema involves the center of the macula, the risk of moderate vision loss is increased tenfold compared to no involvement of the center of the macula [7]. The current treatment options for diabetic retinopathy and DME such as laser photocoagulation, intravitreal anti-vascular endothelial growth factor (VEGF) agent injections, or pars plana vitrectomy basically target the pathological changes based on the vascular abnormalities. Thus, most patients have already had moderate to severe vision loss at the commencement of the treatment. A recent definition of diabetic retinopathy by the American Diabetes Association is that it is a tissue-specific neurovascular impairment of multiple cell types in the retina in both type 1 and type 2 diabetes [8]. The different cell types include neurons, glial cells, and vascular cells. These cells constitute the NVUs, and the interdependence of these cells is associated with maintaining the homeostasis of the retinal environment under healthy conditions.

Among the cells that make up the NVUs, the neuronal cells are more susceptible than the other cells under diabetic conditions. In fact, most studies suggest that neuronal changes precede the vascular changes in the clinical findings of the retinas of diabetic patients [9,10,11,12,13,14,15,16,17,18]. Because neuronal cell death is irreversible and directly related to vision loss, neuroprotection is required to prevent the development and the progression of diabetic neuropathy in the retina at the early stage of diabetic retinas [19,20]. This review focuses on the pathological changes of the NVUs and possible therapeutic options for neuroprotection and vasoprotection of diabetic retinopathy.

## 2. Physiology and Pathology of NVU in Diabetic Retinopathy

### 2.1. Physiology of NVU in Normal Retina

The NVUs are composed of neurons (ganglion cells, amacrine cells, bipolar cells, and horizontal cells), glial cells (Müller cells, astrocytes, and microglia) and vascular cells (endothelial cells and pericytes) [19,20,21]. The anatomical and functional interdependence of these cells help to maintain retinal homeostasis and function under healthy conditions.

The blood–retinal barrier (BRB) is composed of the endothelial cells, pericytes, basement membrane (BM), and glial processes. Physiologically, the retinal blood circulation is controlled by cells belonging to the NVUs [22]. Functional hyperemia is one of the representative interdependences of the NVUs. Briefly, activated neurons release glutamate and K^+^ from their synapses. Astrocytes uptake the K^+^ by inwardly rectifying K+ channels and the glutamate binds to metabotropic glutamate receptors, followed by increasing inositol triphosphate synthesis and increasing the Ca^2+^ concentration in astrocytes [22]. High concentrations of Ca^2+^ produce arachidonic acid (AA) from the membrane phospholipids through the activation of phospholipase A_2_. AA is metabolized to prostaglandin E_2_, prostaglandin I_2_, and epoxyeicosatrienoic acids, which result in vasodilation. When AA is converted to 20-hydroxy-eicostateraenoic acid, prostaglandin F_2_, and thromboxane A_2_, vasoconstriction is induced. Müller cells are the unique and major glial cells in the retina. In the normal retina, light stimulation triggers an increase of retinal blood flow because photoreceptors require more sources of energy. Müller cells supply oxygen and other nutrients such as lactate to the photoreceptors by regulating retinal blood flow and metabolizing glucose to lactate for the photoreceptors [23]. Although the precise mechanism for the maintenance of the BRB is not known, Müller cells can release pigment epithelium-derived factor (PEDF) and thrombospondin-1, which result in a tightening of the tight junctions of the endothelial cells and maintain the BRB [24]. PEDF released from Müller cells affects not only the BRB but also the retinal neurons. Under hypoxic conditions, PEDF, vascular endothelial growth factor (VEGF), and interleukin-6 (IL-6) are upregulated in Müller cells, which protects the retinal ganglion cells from ischemic insults [25]. These findings indicate the existence of interdependence between the neurons and glial cells in the retina.

Astrocytes and Müller cells join in the construction of the BRB, and they regulate the BRB function because their endfeet cover the surrounding blood vessels. For example, an electron microscopic 3D reconstruction of the brain demonstrated that over 99% of the microvessels in the brain were covered with the endfeet of the astrocytes [26]. Thus, it is not difficult to imagine that astrocytes and Müller cells are one of the components of the NVUs. The classical BRB is composed of the horizontal adhesion of the endothelial cells with tight junction proteins. including zonula occludens-1 (ZO-1), occludin, and claudin-5 [27]. These tight junction proteins have decreased expression under diabetic conditions, which results in an increase of vascular permeability [27]. The BM surround the vessels and the endfeet of the astrocytes are connected to the BM. Laminin 1 and 5 and perlecan in the BM are the ligands of the integrin receptors of the astrocytes, and they are connected to the astrocyte-specific receptors, dystroglycan [28]. Thus, astrocytes and endothelial cells can make the vertical adhesions as the second barrier of the BRB. In fact, over 50% of functional hyperemia depends on astrocytes in the brain. In the retina, the Müller cells may share the role of functional hyperemia in the retina with astrocytes.

Astrocytes are not only involved in functional hyperemia but are also involved in the astrocyte–neuron lactate shuttle [29]. Briefly, excited neurons release glutamate and astrocytes reuptake the glutamate via the glutamate transporter. The glutamate activates the glycolysis system in the astrocytes which results in the induction of the anti-oxidant responses through the pentose phosphate pathway and glutathione followed by the protection of neurons.

Microglia are one of the components of the NVUs. Microglia have two activated phenotypes, the proinflammatory (M1) state and the anti-inflammatory (M2) state [30]. In the healthy condition, neurons release fractalkine (CX3C ligand 1). Fractalkine binds to its CX3C chemokine receptor 1 on the M2 state of microglia, which results in regulating microglia repopulation and maintaining the retinal homeostasis [31]. The M1 state of microglia secretes proinflammatory cytokines such as interleukin-6 (IL-6), IL-8, IL-1β, and tumor necrosis factor-α (TNF-α). The M2 state of microglia releases the anti-inflammatory cytokines such as IL-4, IL-10, IL-13, or transforming growth factor-β [30]. In diabetes, the anti-inflammatory state (M2) shifts to the proinflammatory state (M1) and the shift precedes the neuronal cell death in the diabetic retina [32,33,34]. Activated astrocytes are known to specifically produce and release the signaling protein, S100B. The target of S100B is the receptor of the advanced glycation end-products (RAGE) of microglia [35]. In fact, the supplementation of S100B in the culture media leads to abnormally activated microglia via RAGE activation [35].

### 2.2. Pathology of NVU in Diabetic Retinopathy

In the diabetic retina, the interdependence between different cell types consisting of the NVUs is impeded by many stresses, including the polyol pathway [36], the hexosamine pathway [37], oxidative stress [38,39], protein kinase C (PKC) [40,41], advanced glycation-end products (AGEs) [42], and inflammation [43]. The hypothetic scheme of the neurovascular impairments provoked by these stresses in diabetic retinopathy is shown in Figure 1. These stresses impair the function of the NVUs independently or mutually under hyperglycemic stress. Most diabetic stress-related pathways, including reductive stress and the enediol pathway, can culminate in oxidative stress [37].

Glial dysfunction may occur earlier than neuronal abnormalities because glial cells monitor the pathological changes of the vascular environment, such as chronic hyperglycemia, leukocyte adhesions, inflammatory changes, and metabolic changes, which result in glial dysfunction [44,45]. For example, in the early stage of diabetes, the glial fibrillary acidic protein (GFAP) expression is increased in Müller cells [46] and decreased in astrocytes [47]. Müller cell swelling is observed in the early diabetic retinopathy [48], probably because the aquaporins and Kir4.1 channels are increased in Müller cells [49]. These changes lead to the production of delta-like proteins, VEGF-A, and angiopoietin-related protein 4, followed by the promotion of angiogenesis and vascular permeability [49]. However, GFAP expression is a sign of reactive gliosis, but it has not been determined whether GFAP expression is a marker of the first responses of glial cells under stress. Taylor et al. demonstrated that the upregulation of GFAP in Müller cells under various stresses was preceded by the expression of homeostatic regulatory proteins and apoptotic cell death [50]. They concluded that the loss of Müller cell function is not a consequence of gliosis [50]. Thus, pre-gliotic events of the glial cells under diabetic stress should be determined to understand the early pathological changes in neurovascular abnormalities of diabetic retinas. For example, an earlier study showed that Müller cell activation contributed to the neuroprotection through extracellular signal-related kinase 1/2 (ERK1/2) activation under hyperglycemic conditions and concludes [51]. The authors concluded that activated Müller cells do not always lead to neuronal cell death in diabetic retinas [51]. On the other hand, another study using VEGF knockout in Müller cells showed that Müller-cell-derived VEGF is essential for vascular leakage and retinal inflammation in diabetic retinopathy [52]. Further studies are needed to determine the role of glial cells in the pathogenesis of early diabetic retinopathy.

Retinal pericytes play the functional role of maintaining the homeostasis of the retinal vascular system and the formation of the BRB [53,54]. During vascular formation, the endothelial cells release platelet-derived growth factor subunit B (PDGFB), resulting in the recruitment of pericytes to make the proper the BRB formation [54] (Figure 2). Pericytes control the angiopoietin-2 and VEGF receptor 2 (VEGFR2) expressions in endothelial cells in a Forkhead box protein O1-dependent manner, which results in regulating the VEGFA signaling [54]. Chronic hyperglycemia reduces the PDGE receptor signaling, resulting in pericyte apoptosis via protein kinase Cδ (PKCδ) activation and increased Src homology 2 domain-containing phosphatase-1 (SHP1) expression [55,56] (Figure 2). As a result, a pericyte loss via PDGFR signaling reduction results in promoting VEGFA signaling and vascular permeability. In addition, in experimental diabetic retinopathy, the pericyte loss occurs before the endothelial cell loss [57]. Although it is a personal hypothesis, pericyte loss could be a trigger of microaneurysm formation (Figure 2) because the loss of pericytes may cause a decrease in the local extracellular matrix (ECM) production, which results in the BM disruption. In addition, an increase of VEGFA signaling may cause endothelial cell migration followed by microaneurysm formation (Figure 2).

Oxidative stress plays a pivotal role in the pathogenesis and the progression of chronic retinal diseases, such as diabetic retinopathy or retinitis pigmentosa [38,39,61,62]. During the process of the NVU impairments in diabetic retinopathy, reactive oxygen species (ROS) including superoxide anion and hydrogen peroxide are generated because of the imbalance of free radical formation and removal [38,39]. Most pathological pathways, including the polyol pathway, the hexosamine pathway, the AGE pathway, and the PKC pathway, can culminate in oxidative stress [37,38]. For example, in the polyol pathway, aldose reductase transfers glucose into sorbitol and sorbitol dehydrogenase converts sorbitol into fructose [37,38]. The fructose is converted to fructose-3-phosphate and 3-deoxyglucosone, both of which are the precursors of the AGEs [38]. During this process, an antioxidant molecule, nicotinamide adenine dinucleotide phosphate (NAPDH), is excessively consumed, which results in reducing the synthesis of glutathione. In addition, nicotinamide adenine dinucleotide (NADH) is increased, followed by PKC activation. Taken together, the polyol pathway contributes to the production of ROS [37,38]. The excessive accumulation of ROS causes inflammation, NVU impairments, and cell death. The alterations of several transcription factor signalings, such as activator protein-1 (AP-1), p53, or nuclear factor-kappa B (NF-κB), are induced by the oxidative stress, which can result in inflammation and cell death in diabetic retinopathy [39]. These epigenetic changes induced by oxidative stress are important but are not described in detail in this review because that is beyond the scope of this review.

Inflammation is one of the pathological factors that can develop through the whole process in the progression of diabetic retinopathy [63]. For example, DME is one of the vision-threatening changes in the macula of diabetic patients and 4% of diabetic patients have clinically significant macular edema [64]. DME can develop even in the early stages of diabetic retinopathy. The precise mechanism for the development of DME is still unknown but anti-VEGF therapies are the gold standard for the treatment of DME [65,66,67,68,69,70,71,72,73,74]. Thus, VEGF is highly likely to be associated with the pathogenesis of DME. However, some patients have refractory DME even after repeated anti-VEGF intravitreal injections. Not only VEGF, but also other inflammatory cytokines such as IL-6, IL-8, interferon-induced protein-10 (IP-10), monocyte chemotactic protein-1 (MCP-1), and intercellular adhesion molecule-1 (ICAM-1) are involved in the development of DME [75,76]. Anti-VEGF agents cannot reduce the level of the other cytokines in the vitreous, but intravitreal triamcinolone acetonide injections can reduce these inflammatory cytokines, including IL-6, IP-10, MCP-1, and VEGF, significantly in eyes with DME [77]. In fact, switching from anti-VEGF to triamcinolone acetonide injections led to significant therapeutic effects in eyes with refractory DME treated with repeated anti-VEGF injections [78]. Sub-Tenon triamcinolone acetonide injections are also known to be effective in the treatment of DME [79,80,81,82]. Because steroids are strong anti-inflammatory agents, inflammation is involved in the development of DME.

The basic scheme of the mechanism of NLRP3 inflammasome activation is shown in Figure 3. TLR4 expression is induced by the exposure of retinal endothelial cells to high glucose [83]. TLR4 is associated with the pathogenesis of streptozotocin-induced diabetic rats, but inflammatory cytokines such as TNF-α or IL-1β are induced in both a MyD88-dependent and MyD88-independent manner [84]. In addition, the inhibition of TLR4 reduces inflammation and retinal ganglion cell apoptosis under high glucose exposure [85]. The results of another study indicate that TLR2 and 4, NF-κB, TNF-α, and IL-8 are upregulated in retinal ganglion cells under high glucose conditions [86]. Activated microglia and Müller cells release proinflammatory cytokines including IL-1β and TNF-α in diabetic retinopathy [87,88]. Furthermore, pyroptosis, an inflammatory cell death mediated by gasdermin D, is involved in the NVU impairments in diabetic retinopathy [89]. Taken together, anti-inflammatory therapies may be one of the options for preventing the progression of diabetic retinopathy.

Chronic hyperglycemia causes glycation between glucose and amino acids, which is called the Maillard reaction, and the nonenzymatic glycation reactions give rise to additional irreversible covalently bound Amadori rearrangement products termed the AGEs [90]. A thickening of the vascular BM is observed in diabetic retinopathy which results in an increased vascular permeability [59,60]. Aberrant crosslinks between AGEs and ECM components including fibronectin, laminin, and collagen IV lead to an increase in the BM thickness [91]. Furthermore, the binding of AGEs to RAGE expressed in various types of cells, including glial cells, vascular cells, and neurons, results in activating multiple cellular signaling pathways in diabetic retinopathy. The hypothetic scheme of the cellular events involved in AGEs is shown in Figure 4.

We used a 3D collagen gel culture system and cultured rat retinal explants in 10 μg/mL glucose-AGE-bovine serum albumin (BSA) incubation media [92]. In glucose-AGE-BSA-exposed retinas, the number of cells undergoing apoptosis in the ganglion cell layer (GCL) was significantly greater and the regenerating neurites were fewer than those in retinas cultured in control media [92]. Between 1 and 120 μg/mL of AGEs, which is equal to 4–480 μg/mL of glycated BSA, is known to circulate in diabetic patients [93]. Because low concentrations of glucose-AGE-BSA have toxic effects on retinal neuronal cells, the accumulation of AGEs can induce retinal degeneration in diabetic retinas. Furthermore, in 10 μg/mL glycolaldehyde-AGE-BSA and 10 μg/mL glyceraldehyde AGE-BSA-incubated retinas, the number of apoptotic cells was greater than in glucose-AGE-BSA incubated retinas [92]. Thus, glycolaldehyde and glyceraldehyde are more toxic than glucose-AGE.

When retinas are incubated in 100 μg/mL glucose AGE-BSA, the numbers of apoptosis and NF-κB immunopositive cells are significantly increased, and in neurotrophin-4 (NT-4)-incubated retinas cultured in AGE-BSA, these numbers were significantly decreased compared with retinas cultured in control media [94]. Thus, the toxic effects of AGEs against retinal neurons are corelated with the expression of NF-κB. Furthermore, 100 ng/mL NT-4, 100 ng/mL hepatocyte growth factor (HGF), 100 ng/mL glial cell line-derived neurotrophic factor (GDNF), and 100 μM tauroursodeoxycholic acid (TUDCA) significantly reduced the numbers of apoptosis and NF-κB expression [94] (Figure 4). Because TUDCA shows protective effects against AGE-induced retinal neuronal cell death, the ER stress-related cell death pathways are involved in AGE-induced retinal neuronal apoptosis (Figure 4).

**Figure 4 ijerph-19-00439-f004:**
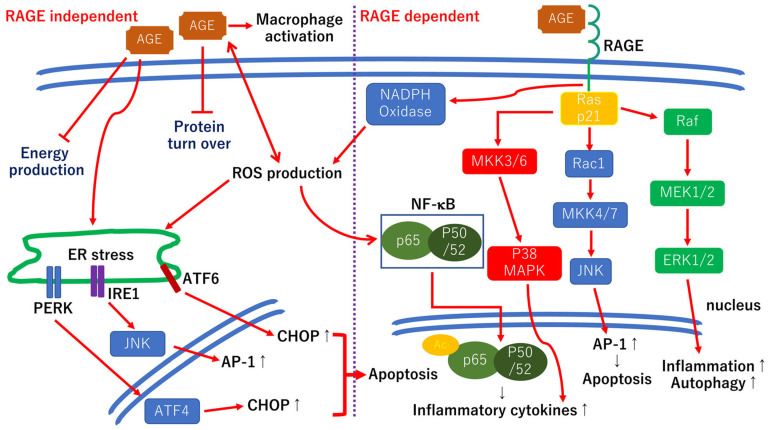
Hypothetic scheme for both RAGE-dependent and RAGE-independent cellular signaling pathways. The AGE-RAGE-dependent signaling pathways activate various cellular downstream pathways including the NADPH–NF–κB axis and the Ras–MAPK axis, which result in inflammatory reactions [95]. The Ras–ERK1/2 axis is also involved in the inflammatory reactions. The Ras-c-Jun N-terminal kinase (JNK) axis is associated with apoptotic cell death. The RAGE-independent pathway is involved in the BM thickening by disturbing the ECM protein turn-over rate [59,60]. The AGEs upregulate the CCAAT-enhancer-binding protein homologous protein (CHOP) levels and induce endoplasmic reticulum (ER) stress-related cell death in retinal pericytes [96]. In addition, AGEs, and Amadori products become the source of the ROS. Raf: rapidly accelerated fibrosarcoma; Rac1: PAS-related C3 botulinus toxin substrate 1; MEK1/2: MAPK ERK kinase 1/2; MKK4/7: mitogen-activated protein kinase 4/7; MAPK: mitogen-activated protein kinase; PERK: protein kinase-like ER eukaryotic initiation factor-2 alpha kinase; IRE1: inositol-requiring ER-to-nucleus signaling protein 1; ATF4, 6: activating transcription factor-4, -6; Ac: acetylation.

Similarly, the phosphorylated-JNK and p38 expressions were significantly increased in 100 μg/mL glucose AGE-BSA incubate retinas compared to retinas cultured in normal medium [97] (Figure 4). Thus, the AGE-RAGE pathways are, in part, associated with neuronal cell death in AGE-induced cell death in retinas.

However, in AGE-exposed retinas incubated with the RAGE inhibitor, the number of apoptotic cells was partly reduced, and the number of neurites was partly increased compared to retinas cultured in normal medium, and the neuroprotective and regenerative effects of the RAGE inhibitor are weaker than that of other neurotrophic factors [98]. Thus, the AGEs induced retinal neuronal apoptosis in both RAGE-dependent and RAGE-independent manner [98], and single RAGE inhibition therapy may not be sufficient for protecting retinal neurons in diabetic retinas (Figure 4).

## 3. Therapeutic Options for NVU Impairments in Diabetic Retinopathy

Evidence for the therapeutic options for diabetic retinopathy has been greatly increased recently. Most researchers have been focusing on the treatments for the NVU impairments of diabetic retinopathy, i.e., treatments for both vascular and neuronal abnormalities in diabetic retinopathy.

### 3.1. Antioxidants

There are mainly two types of antioxidants, the flavonoids and carotenoids, which reduce oxidative stress in diabetic retinopathy. Because most of diabetic stress-induced pathways may culminate in oxidative stress [37], antioxidants may be one of the promising therapeutic options for preventing the progression of early diabetic retinopathy.

One of representative flavonoids is curcumin. Administration of curcumin in diabetic rats improved both the vascular abnormalities and neuronal abnormalities, i.e., reduced the VEGF and TNF-α, protection of capillary BM thickening, oxidative stress reduction, and protection of the inner retinal damage [99,100,101]. A recent clinical trial indicates that curcumin slightly reduces the levels of proinflammatory cytokines and soluble mediators in the vitreous of patients with proliferative diabetic retinopathy (*n* = 28) [102]. Representative carotenoids are lutein and zeaxanthin. Both clinical and experimental studies demonstrated their therapeutic effects on diabetic retinopathy. In experimental diabetes, lutein and zeaxanthin reduced the production of ROS, inflammatory cytokines, vascular abnormalities, retinal apoptosis, and protect the visual function [103,104]. In clinical trials, lutein and zeaxanthin improved the macular function in eyes with DME [105] and improved the retinal response density in type 2 diabetic patients [106]. However, in the former study, n values were small, i.e., 30 in the control, 30 in the diabetic retinopathy, and 30 in the diabetic retinopathy with treatment, and the follow-up period was short (3 months) [105]. The latter study is a retrospective study with no control group [106]. Thus, further randomized large control trials are required to demonstrate the effect of lutein and zeaxanthin for treatment of diabetic retinopathy.

An oxygen free radical scavenger, calcium dobesilate (CaD), has been used as a vasoprotective agent because CaD inhibits vascular permeability and reduces vascular leakage by the downregulation of VEGF, ICAM-1, and proinflammatory cytokines [107,108,109]. In addition, CaD improves the electroretinogram (ERG) responses by preventing glutamate transporter downregulation in diabetic mice retinas [110]. Previous clinical trials indicated that in the early stage of diabetic retinopathy, CaD prevented the progression of diabetic retinopathy [111,112]. However, a later clinical trial demonstrated that CaD did not reduce the development of clinically significant macular edema [113]. Thus, further studies are needed to demonstrate the effect of CaD in preventing the development and the progression of diabetic retinopathy. Now, a single-blind, multicenter, 24-armed cluster-randomized, controlled trial is performing to evaluate the effect of CaD on preventing the progression of diabetic retinopathy (NCT04283162) [114].

Nuclear factor erythroid 2-related factor 2 (Nrf2) regulates many antioxidant genes as a master transcription factor, and Nrf2 activation protects tissues from oxidative stress [115]. In fact, the Nrf2 activator dh404 reduces the oxidative stress and inflammatory mediators including VEGF and improves the increase of the vascular leakage [116]. An intracellular inhibitor, Kelch-like ECH-associated protein 1 (Keap1), mediates the translocation and transcriptional activity of Nrf2 [115]. In diabetic retinas, the Keap1 levels are increased, which results in decreasing the nuclear translocation and transcriptional activity of Nrf2 [117]. Long noncoding RNAs (LncRNAs), classes of transcribed RNA molecules with more than 200 nucleotides, regulate gene expression by bringing the transcription factors to the transcription start site and interacting with DNA at the gene promotor site [118]. Metastasis-associated lung adenocarcinoma transcript 1 (*MALAT1*) conserves LncRNAs that are involved in inflammation, cell death, and angiogenesis in diabetic retinopathy [119]. The results of a recent study indicated that LncRNA *MALT1* facilitated its transcription by increasing the binding of the transcription factor at the *Keap1* promoter, which results in preventing a translocation of the master regulator Nrf2 and losing the antioxidant defense in diabetic retinopathy [120]. Thus, inhibition of LncRNA MALT1 may become one of the therapeutic options for preventing the development and the progression of diabetic retinopathy.

### 3.2. Anti-Inflammatory Agents

IL-1β, a pivotal inflammatory cytokine, plays a critical role in the pathogenesis of diabetic retinopathy (Figure 3) [121]. In addition, ICAM-1, IL-6, IL-8, MCP-1, VEGF, and IP-10 are associated with the pathogenesis of diabetic retinopathy [75,76,77]. Although TNF-α is also associated with the pathogenesis of diabetic retinopathy [121], TNF-α antagonist, etanercept, adalimumab, and infliximab fail to improve refractory DME [122,123]. Thus, other anti-inflammatory agents including triamcinolone acetonide [78,79,80,81,82] may be better than anti-TNF-α agents.

Suppressor of cytokine signaling (SOCS) 1 is one of the family members of the SOCS, intracellular cytokines inducible protein. SOCS1 is a negative regulator of IL-6 and TNF-α and regulates cytokine receptor signaling by a negative feedback loop [124]. SOCS3 is the suppressor of the intrinsic neuroprotective and regenerative pathway, the Janus kinase–signal transducers and activators of transcription pathway [125]. The results of a recent study found that the topical instillation of SOCS1-derived peptide in diabetic mice reduced the activated microglia, proinflammatory cytokines including IL-1β and IL-6, and improved the abnormal retinal function and vascular leakage [126].

Microglia play a pivotal role in the inflammatory responses in diabetic retinopathy [30]. Minocycline is one of the tetracycline antibiotics, and it reduces the inflammatory cytokines, including IL-1β, by inhibiting microglia activation in diabetic retinopathy [127]. Minocycline rescues retinal neuronal cell death by reducing caspase-3 activation and abnormal histone methylation levels, which are induced by microglia activation in the diabetic rat retinas [128]. However, the results of a recent study indicated that minocycline partially improved diabetic retinopathy in both type 1 and type 2 mice models, but was not better than the currently available therapies, insulin, or pioglitazone [129]. Thus, minocycline may have limited use for the treatment of diabetic retinopathy.

### 3.3. Neurotrophic Factors and Others

BDNF is one of the nerve growth factor family members and is associated with the pathogenesis of diabetic retinopathy and one of the therapeutic options. The BDNF level in the serum and the aqueous humor in patients with diabetes is significantly reduced [130], which corresponded with the results of animal models [131]. Thus, the serum level of BDNF can be an indicator of the progression of diabetic retinopathy [132]. BDNF and NT-4 bind to the same high affinity receptor, trkB, and they regulate the intrinsic survival and regenerative pathways [133]. Exogenous BDNF protects retinal neurons [134] and facilitates neurite regeneration under high glucose conditions [135]. In our previous study, the regenerative effect of NT-4 appeared to be better than that of BDNF in retinas cultured in high glucose medium [135]. Another study indicated that an overdose of BDNF can cause inflammation of the retina [134]. Thus, the optimal concentration of BDNF should be found for the treatment of diabetic retinopathy. Because the half-life of the BDNF protein is short, several nanoparticles delivery systems have been recently developed [136,137].

PEDF, an intrinsic antioxidant, is secreted by various retinal cells, including the retinal pigment epithelial cells, and it has two actions, viz., vasoprotection and neuroprotection. In diabetic animal models, intravitreal injection of PEDF reduces retinal neuronal damage and VEGF expression as an antioxidant [138]. Furthermore, topical instillation of PEDF reduces retinal ganglion cell death, Müller cell activation, and vascular leakage in diabetic mice retinas [139]. Recently, PEDF-derived peptides with angiogenic inhibitory activities have been developed for the translation of the PEDF treatment into clinical practice [140].

Somatostatin has both antiangiogenic and neuroprotective effects and it acts through somatostatin receptor 2. Topical administration of somatostatin ameliorates neuronal degeneration in diabetic murine models [141]. However, in a multicenter, randomized controlled clinical trial, the EUROCONDOR study, somatostatin eye drops did not protect the retinal neurons but reduced the progression of pre-existing neuronal dysfunction in subpopulations of retinal neurons [142]. In a clinical trial, multifocal electroretinography (ERG) was used to evaluate the primary end point. Other methods such as microperimetry may be better to evaluate the neuroprotective effects of somatostatin [142].

Glucagon-like peptide-1 (GLP-1) and GLP-1 receptors are expressed in the retina [143], and their expression is decreased in diabetic patients [144]. Topical administration of GLP-1 receptor agonists ameliorates the retinal neurodegeneration in diabetic animal models [145]. However, in the LEADER trial, liraglutide, a GLP-1 analogue, did not prevent the progression of diabetic retinopathy. Relatively higher rates of retinopathy events were observed with liraglutide than with the placebo [146]. Furthermore, in the SUSTAIN-6 trial, semaglutide, another GLP-1 analogue, exacerbated the development of diabetic retinopathy more with semaglutide than with the placebo [147]. Taken together, GLP-1 analogues are not effective in treating diabetic retinopathy.

Although it is not a neurotrophic factor, fenofibrate may be an ideal drug to prevent the development and the progression of diabetic retinopathy. Fenofibrate, a peroxisome proliferator-activated receptor alpha (PPARα), is used as a drug to treat hyperlipidemia worldwide. Two randomized clinical trials found that fenofibrate significantly reduced the progression of diabetic retinopathy [148,149] (*n* = 9795 and 10,251, respectively; Table 1). Fenofibrate reduced overexpression of the ECM components, fibronectin and collagen IV, in endothelial cells [150] and retinal pigment endothelial cells [151], which resulted in reducing the BRB leakage [151,152]. Chen et al. reported that PPARα agonists reduced the overexpression of MCP-1, ICAM-1, and VEGF, and inhibited the hypoxia-inducible factor-1 and NF-κB. These changes resulted in an ameliorating vascular leakage in diabetic retinas [152]. Furthermore, fenofibrate reduced glial activation and retinal neuronal apoptosis, followed by improving the ERG parameters [153]. Because of having both vasoprotective and neuroprotective effects, fenofibrate is an ideal drug to target the NVU impairments of early diabetic retinopathy.

DME is one of the common causes of vision decrease in the early stage of diabetic retinopathy. Although several therapeutic procedures are effective for improving DME, anti-VEGF therapies have become the gold standard for the management of DME [63,64,65,66,67,68,69,70,71,72]. However, VEGF is an endogenous neurotrophic factor for retinal ganglion cells [135,154]. Furthermore, VEGF expressed in vascular cells is considered to play physiological roles for the retinal vascular system. A Japanese survey indicated that a new and strong anti-VEGF agent, brolucizumab, has various side effects including intraocular inflammation (9.4%), retinal vasculitis (3.1%), and retinal vascular occlusion (1.6%) [155]. The direction of strengthening VEGF inhibition may be a limitation for the complete management of diabetic retinopathy. The early intervention of preventing the development of the NVU dysfunction is one of the options for resolving this limitation.

## 4. Conclusions

Diabetic retinopathy is a disease accompanied by NVU impairments, and all the pathological changes, including to vascular cells, glial cells, and neuronal cells, should be ameliorated so to prevent the development and progression of diabetic retinopathy. Neuronal cell death is an irreversible event which directly results in vision loss. Thus, neuroprotective therapies for diabetic retinopathy should be established. However, the target of ideal therapies for a neurovascular disease such as diabetic retinopathy should focus on the NVU impairments; thus, both vasoprotective and neuroprotective therapies are required for preventing vision loss in patients with diabetic retinopathy. Understanding the NVU impairments and establishing therapies for neurovascular protection should be helpful for establishing new ways of managing diabetic retinopathy in the future.

## Figures and Tables

**Figure 1 ijerph-19-00439-f001:**
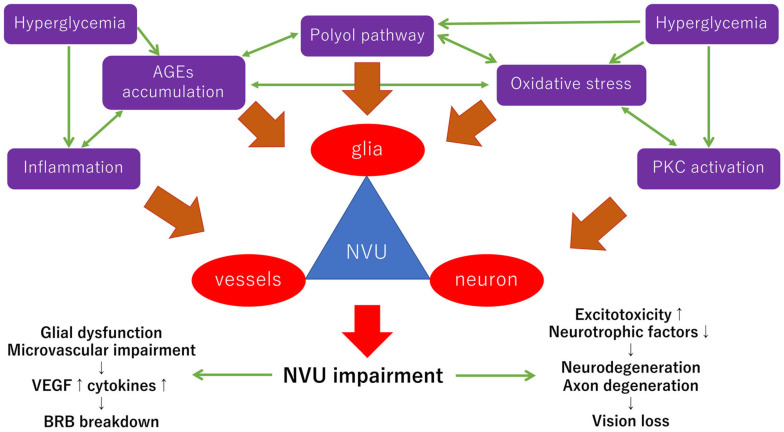
Hypothetical scheme of NVU impairments in early diabetic retinopathy. The NVUs are composed of neurons, glial cells, and vascular cells. Chronic hyperglycemia is a trigger for the impairment of the NVUs. Chronic hyperglycemia causes several stresses, including AGEs accumulation, oxidative stress, inflammation, activation of the polyol pathway, and PKC activation. These stresses lead to glial dysfunction and microvascular impairments, which result in increasing the expression of VEGF and inflammatory cytokines in the eye followed by the BRB breakdown. Neurons and their axons are degenerated by excitotoxicity, neurotrophic factors reduction, which is directly related to vision loss. Most neuronal abnormalities may be independent of the vascular abnormalities in the early stage of diabetes. Not all stresses such as the hexosamine pathways are included in the figure because of the limited space, and the hexosamine pathways may be involved in the late stage of the pathogenesis of diabetic retinopathy [44]. NVU: neurovascular unit; AGEs: advanced glycation end-products; PKC: protein kinase C; VEGF: vascular endothelial growth factor; BRB: blood–retinal barrier.

**Figure 2 ijerph-19-00439-f002:**
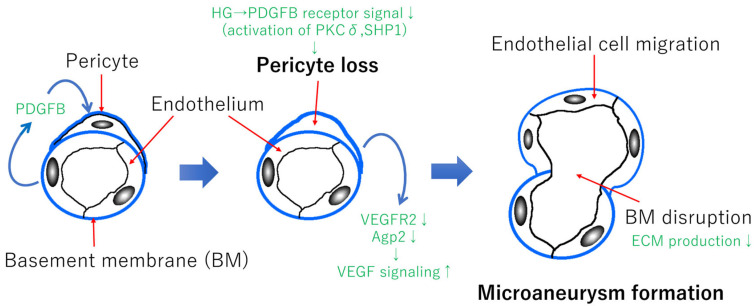
Hypothetical scheme of microaneurysm formation. PDGFB released from endothelial cells recruit pericytes and make the BRB formation during the process of retinal vascular development [53,54]. Chronic hyperglycemia reduces the PDGFB receptor signaling followed by pericyte apoptosis via activation of PKCδ and increased expression of SHP1 [55,56]. These molecular events lead to pericyte loss, which results in reducing VEGFR2 and angiopoietin-2 signaling [55,56]. As a result, VEGFA signaling is increased, which may cause endothelial cell migration. The BM in the retinal vessels is composed of ECM components, including fibronectin, collagen IV, and laminin derived from endothelial cells and pericytes [58]. In general, chronic hyperglycemia or high glucose conditions promote the synthesis of the ECM components, which results in an increase of vascular permeability [58,59,60]. However, pericyte loss may cause a disruption of the BM locally because of a reduction of ECM production only in the area of the pericyte loss. Taken together, microaneurysms may be formed, as shown in Figure 2. PDGFB: platelet-derived growth factor B; HG: hyperglycemia; PKCδ: protein kinase Cδ; SHP1: Src homology 2 domain-containing phosphatase-1; VEGFR2: vascular endothelial growth factor receptor 2; Agp2: angiopoietin-2; ECM: extracellular matrix.

**Figure 3 ijerph-19-00439-f003:**
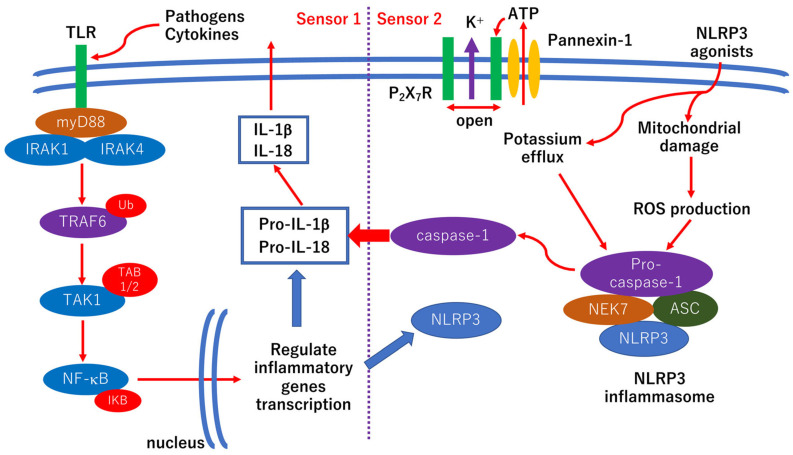
Simple scheme of the mechanisms of Nod-like receptor family pyrin domain containing 3 (NLRP3) inflammasome activation. For a better understanding of the mechanism of NLRP3 inflammasome activation, the factors described in the scheme are based on the general inflammatory reaction. Thus, not all factors are demonstrated in diabetic models. Pathogens or cytokines such as tumor necrosis factor-α (TNF-α) and IL-1β bind to the Toll-like receptor (TLR), leading to the activation of NF-κB signaling. NF-κB regulates the transcription of the inflammatory genes including pro-IL-1β, pro-IL-18, and NLRP3 (Sensor 1). NLRP3 agonists, including damage associated molecular pattern molecules (DAMPs) and pathogen-associated molecular pattern molecules (PAMPs), cause mitochondrial damage, which results in ROS production. In addition, NLRP3 agonists induce potassium efflux via the K channel linked with P_2_X_7_ receptors (P_2_X_7_R). These changes lead to NLRP3 inflammasome activation followed by caspase-1 activation. This results in the cleavage of pro-IL-1β and pro-IL-18. After that, mature IL-1β and IL-18 are released externally, and an inflammatory reaction is induced. Excessive potassium efflux and activation of caspase-1 may cause pyropotsis by a cleavage of gasdermin D. MyD88: myeloid differentiation primary response gene 88; IRAK: IL-1 receptor-associated kinase; TRAF6: TNF receptor-associated factor 6; Ub: ubiquitin; TAK1: TGF-β-activated kinase 1; TAB1/2: TAK1-binding protein 1/2; IκB: inhibitor of κB; MEK7: MAPK ERK kinase 7; ASC: apoptosis-associated speck-like protein containing a CARD.

**Table 1 ijerph-19-00439-t001:** Summary of treatment options for early diabetic retinopathy.

Treatment Options	Compounds	References	Remarks
Antioxidants	Curcumin (flavonoids)	[99,100,101]	Basic studies
		[102]	Clinical study (*n* = 28) (positive)
	Lutein (carotenoids) Zeaxanthin (carotenoids)	[103,104]	Basic studies
		[105]	Clinical study (*n* = 30 × 3 groups, 3 months) (positive)
		[106]	Clinical study (*n* = 60, 120 eyes, retrospective) (positive)
	CaD (free radical scavenger)	[107,108,109,110]	Basic studies
		[111]	Clinical study (*n* = 41, randomized) (positive)
		[112]	Clinical study (*n* = 194, randomized double-blind) (positive)
		[113]	Clinical study (*n* = 635, randomized double-blind, placebo-controlled, multicenter) (negative)
	Nrf2 (master transcriptional factor)	[117,120]	Basic studies
Anti-inflammatory agents	TNF-α antagonists	[122,123]	Clinical studies (negative for DME)
	Triamcinolone acetonide (sub-Tenon) *	[78,79,80,81,82]	Clinical studies (positive for DME)
	SOCS1 (cytokine-inducible protein)	[126]	Basic study
	Minocycline	[127,128,129]	Basic studies
Neurotrophic factors	BDNF	[130,132]	Clinical studies (as a biomarker)
		[131,134,135,136,137]	Basic studies
	PEDF	[138,140]	Basic studies
	Somatostatin	[141]	Basic study
		[142]	Clinical study (*n* = 449, randomized, placebo-controlled, phase II–III) (partly positive)
	GLP-1	[144,145]	Basic studies
		[146]	Clinical study (*n* = 9340, r multicenter, double-blind, placebo-controlled trial) (negative)
		[147]	Clinical study (*n* = 3297, randomized, double-blind, placebo-controlled, parallel-group trial) (negative)
Others	Fenofibrate (PPARα)	[150,151,152,153]	Basic studies
		[148]	Clinical study (*n* = 9795, randomized, placebo-controlled, multicenter trial) (positive)
		[149]	Clinical study (*n* = 10,251, randomized placebo-controlled, multicenter trial) (positive)

* In Japan, sustained release steroid agents have not been approved yet. Instead, sub-Tenon triamcinolone acetonide injections are frequently used.

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
