# Peer review of "Neurovascular Impairment and Therapeutic Strategies in Diabetic Retinopathy"

_ijerph, 2021, doi:10.3390/ijerph19010439_

Round 1

Reviewer 1 Report

I have no objections to the section: "Physiology and Pathology of NVU in diabetic retinopathy". However, this article is entitled ,,Therapeutic strategies in dia betic retinopathy”and abot strategies there is only secton 3. Thus, I should not have doubts whether human or animal studies are cited, and I have such doubts through Secton 3. “Therapeutic options for NVU impairments in diabetic retinopathy”, please correct section 3.

In addition, “Two randomized clinical trials found that fenofibrate significantly reduced (line 437) the progression of diabetic retinopathy [144,145].” – Author should comment how many people are in this study and is it enough for fenofibrate to be therapeutic strategies ? ?

Only in conclusion I read about a drug, brolucizumab, please introduce the reader to the basics of its influence, I mean previously in the main text, if it is criticized in Conclusion. More data about  standards for the management of DME should be added.

Author Response

  1. Please correct section 3.

Answer. The section 3 has been updated and added a new table (Table 1) as a reference.

  1. Two randomized clinical trials found that…

Answer. N values are added in the text and more information is included in the new table regarding all clinical trials.

  1. Only in conclusion I read about a drug, brolucizumab….

Answer. The sentences regarding brolucizumab have been transferred to the section 3 and more explanation has been added in the text briefly. However, the topics of DME are extremely wide and beyond the scope of this review. A different review may be required for the detail description regarding the current topics of DME (I will write the different review article regarding DME in the near future).

Reviewer 2 Report

In this review, the author summarized current knowledge about neurovascular units in the retina and their role in the progression of diabetic retinopathy, as well as different therapeutic approaches. I believe that the present manuscript has a comprehensive, interesting and current approach and could be a starting point for a broader analysis for a better understanding of the possible mechanisms involved in diabetic retinopathy. My overall impression is that this good review would be of great interest to researchers working in the field of diabetic retinopathy and beyond.

However, I have some suggestions that could improve the manuscript:

  • Section 2 should include subsections that distinguish between physiology and pathology.
  • Figure 3 is very general for a review of these characteristics. It should be based on studies done on cells involved in diabetic retinopathy.
  • Finally, it would be advisable to use tables that summarize the different therapeutic approaches.

Author Response

  1. Section 2 should include subsections…

Answer. The subsections, “Physiology of NVU in normal retina” and “Pathology of NVU in diabetic retinopathy” has been inserted in the section.

  1. Figure 3 is very general for a review….

Answer. “For a better understanding of the mechanism of NLRP3 inflammasome activation, the factors described in the scheme is based on the general inflammatory reaction. Thus, not all factors are demonstrated in diabetic models.”

The explanation described above has been inserted in the figure legend.

  1. It would be advisable to use tables that summarize the different therapeutic approaches.

Answer. I have added the new table in the section 3 and updated the text.

Reviewer 3 Report

Oshitari wrote a very interesting review describing the “Neurovascular impairment and therapeutic strategies in diabetic retinopathy”. The manuscript represents an interesting way to investigate the newest scenarios for neuronal degenerations. I only suggest several minor revisions needed to update the paper:  

  • The manuscript is really clear and complete. I only suggest to widen the bibliography, referring to recent papers about the role of oxidative stress into retinal degenerations. Regarding this, I suggest to add the following references to the manuscript PMID: 34058230 and PMID: 33801777. 
  • Finally, manuscript requires English revisions and typos correction.

Author Response

  1. I suggest to add the following references to the manuscript…

Answer. I have added the suggested references in the text.

  1. Manuscript requires English revision…

Answer. The whole manuscript has been checked again to correct spelling and grammar.

Reviewer 4 Report

The article “Neurovascular impairment and therapeutic strategies in diabetic retinopathy” is very informational. This article will help the scientific community those involved in therapeutic strategies in diabetic retinopathy and people starting their research career in this field.

Highlights:

- The article examines the effects of cytokines and pathways related to neurovascular impairment in early diabetic retinopathy.

- Hypothetical scheme of microaneurysm formation is informative, and the details of the endothelial cell migration and the pathways are great.

- Figure in the Physiology and Pathology section gives the readers overall consolidated pathways and mechanisms of Nod-like receptor family pyrin domain containing 3 (NLRP3) inflammasome activation, tumor necrosis factor-α (TNF-α) and IL-1β bind to the toll-like receptor, activation of NF-κB signaling, etc.

RAGE-dependent and RAGE-independent cellular signaling pathways are well characterized and give the readers more understanding.  

Overall, the article gives in-depth details of the Neurovascular impairment in diabetic retinopathy. Researchers and clinicians will be benifited with this article and aid them to develop therapies for neurovascular protection for diabetic retinopathy. 

Author Response

Thank you for your highest evaluation.

Round 2

Reviewer 1 Report

 Accept in present form.

Reviewer 2 Report

The authors have accepted the reviewers' suggestions and have improved the manuscript sufficiently for it to be published in this journal.